# Osteomyelitis of the Lower Limb: Diagnostic Accuracy of Dual-Energy CT versus MRI

**DOI:** 10.3390/diagnostics13040703

**Published:** 2023-02-13

**Authors:** Giovanni Foti, Chiara Longo, Claudia Sorgato, Eugenio Simone Oliboni, Cristina Mazzi, Leonardo Motta, Giulia Bertoli, Stefania Marocco

**Affiliations:** 1Department of Radiology, IRCCS Sacro Cuore Don Calabria Hospital, Negrar di Valpolicella, 37042 Verona, Italy; 2Department of Diabetology, IRCCS Sacro Cuore Don Calabria Hospital, Negrar di Valpolicella, 37042 Verona, Italy; 3Clinical Research Unit, IRCCS Ospedale Sacro Cuore Don Calabria, Negrar di Valpolicella, 37042 Verona, Italy; 4Department of Infectious, Tropical Diseases and Microbiology, IRCCS Sacro Cuore Don Calabria Hospital, Negrar di Valpolicella, 37042 Verona, Italy

**Keywords:** chronic pain, osteomyelitis, MRI, dual-energy CT, bone marrow edema

## Abstract

Background: MRI is the preferred imaging technique for the identification of osteomyelitis. The key element for diagnosis is the presence of bone marrow edema (BME). Dual-energy CT (DECT) is an alternative tool which is able to identify BME in the lower limb. Purpose: To compare the diagnostic performance of DECT and MRI for osteomyelitis, using clinical, microbiological, and imaging data as reference standards. Materials and Methods: This prospective single-center study enrolled consecutive patients with suspected bone infections undergoing DECT and MRI imaging from December 2020 to June 2022. Four blinded radiologists with various experience levels (range of 3-21 years) evaluated the imaging findings. Osteomyelitis was diagnosed in the presence of BMEs, abscesses, sinus tracts, bone reabsorption, or gaseous elements. The sensitivity, specificity, and AUC values of each method were determined and compared using a multi-reader multi-case analysis. A *p* value < 0.05 was considered significant. Results: In total, 44 study participants (mean age 62.5 years ± 16.5 [SD], 32 men) were evaluated. Osteomyelitis was diagnosed in 32 participants. For the MRI, the mean sensitivity and specificity were 89.1% and 87.5%, while for the DECT they were 89.0% and 72.9%, respectively. The DECT demonstrated a good diagnostic performance (AUC = 0.88), compared with the MRI (AUC = 0.92) (*p* = 0.12). When considering each imaging finding alone, the best accuracy was achieved by considering BME (AUC for DECT 0.85 versus AUC of MRI of 0.93, with *p* = 0.07), followed by the presence of bone erosions (AUC 0.77 for DECT and 0.53 for MRI, with *p* = 0.02). The inter-reader agreement of the DECT (k = 88) was similar to that of the MRI (k = 90). Conclusion: Dual-energy CT demonstrated a good diagnostic performance in detecting osteomyelitis.

## 1. Introduction

Osteomyelitis is an infection of the bone which involves the medullary canal [1]. It may present as an acute or chronic inflammatory process secondary to an infection with pyogenic organisms, including bacteria, fungi, and mycobacteria, and may be associated with chronic pain [2]. Foot localization is the most frequent infection site for diabetic patients, and such infection mostly occurs from the contiguous spread of a soft tissue infection from an adjacent skin ulceration or from a post-operative soft tissue defect [3]. In all other localizations, osteomyelitis may be caused by a hematogenous spread, a spread from a contiguous infected source, or a direct implantation or after surgery [4]. 

Diagnosing osteomyelitis requires a combination of clinical, laboratory, microbiological, and imaging findings [5]. Microbiological diagnosis is based upon the identification of bacteria and the presence of inflammatory cells and osteonecrosis from an uncontaminated sample [6]. Bone biopsies are commonly used for diagnosis, but their diagnostic yield can greatly vary [7]. Even though microbiological identification remains the gold standard for diagnosis, the microbiological accuracy from relevant samples persist to be low, as the culture yields positive results in only 21–28% of the cases [7]. While insufficient material or prior antibiotic therapy may cause false-negative results, false-positive results may arise from contaminants colonizing the skin or wound [1]. 

Radiographs can be inaccurate in the detection of osteomyelitis, with a pooled sensitivity of 54% and specificity of 68% [8,9]. Furthermore, osseous changes may not be radiographically evident for 7–15 days following the onset of osseous infection [9].

MRI is the preferred technique to identify osteomyelitis and to evaluate the extent of the soft tissue infection [10,11], as well as to plan a surgical resection [12]. MRI is able to identify the BME of the involved bone, representing a key element for diagnosing osteomyelitis. In the evaluation of pedal osteomyelitis, MRI has a high sensitivity (90%) and specificity (ranging from 79% to 83%) and it is the preferred technique to evaluate soft tissue abscesses, with a reported sensitivity of 97% and specificity of 77% [8,9,13,14,15]. The primary barriers to the use of MRI are limited access and high costs. 

Dual-energy CT (DECT) has been extensively used for the identification of BME in traumatic and non-traumatic settings [16,17,18,19,20,21]. In particular, in non-traumatic patients, DECT showed a sensitivity of 91% and a specificity of 95% in the identification of BME of the ankle [22]. Moreover, recent data suggest that DECT may be proposed for the identification of osteomyelitis, showing a sensitivity of 81% and a specificity ranging from 73% to 81% [23].

The purpose of our study was to investigate the diagnostic performance of DECT compared to MRI for diagnosing osteomyelitis, using microbiological and biopsy data as reference standards for diagnosis.

## 2. Material and Methods

### 2.1. Participants

This prospective single-center study was conducted in a large tertiary referral hospital after approval by the institutional review board (IRB). Written informed consent was obtained from all participants. Between December 2020 and June 2022, patients presenting to our institution with a suspected osteomyelitis who also underwent both an MRI and a DECT examination within the space of a week were considered for inclusion in the study. A diagnosis of osteomyelitis was confirmed by a comprehensive multidisciplinary assessment based on clinical, microbiological, and imaging features. The exclusion criteria were: oncologic patients, the lack of imaging examinations, or the absence of clinical and microbiological variables. 

### 2.2. Magnetic Resonance Imaging

MRI was performed with a commercially available 1.5-T unit (Magnetom Avanto Fit; Siemens Healthineers, Erlangen, Germany). Standard 4-mm thick T1-weighted turbo spin-echo (TR/TE/FA = 650.0 ms/18.0 ms/150°) sequences were acquired on the axial and coronal planes, and T2-weighted turbo spin-echo (TR/TE = 4300.0 ms/124.0 ms) sequences on the axial and sagittal planes. Furthermore, proton density fat-saturated sequences (TR/TE/FA = 2320.0 ms/39.0 ms/150°) were acquired on the axial and coronal planes for the detection of bone marrow and soft tissue edema. Finally, a 3D isotropic T1-weighted VIBE sequence (TR/TE/FA = 5.9 ms/2.1 ms/FA 10/NEX 2) was acquired on the axial plane after the intravenous administration of gadolinium (Dotarem) and reconstructed on the coronal and sagittal planes.

### 2.3. DECT Protocol

All dual-energy CT exams were performed without the intravenous injection of contrast material. A third-generation scanner was employed (Somatom Definition Force, Siemens Healthineers). A dual-source acquisition technique was used for setting the tube voltages at 80 and 150 kVp with a tin filter. The tube current–time product was set at 1.6:1 (tube A, 220 mAs; tube B, 138 quality reference mAs). Thanks to the implementation of automated attenuation-based tube current modulation (CARE dose 4D; Siemens Healthcare), the radiation burden was similar to that of similar previous studies.

### 2.4. DECT Post-Processing

Soft-tissue kernel (Qr32) 80 kV and 150 kV set images (thickness 0.75 mm; increment 0.6 mm) were transferred to an offline workstation (SyngoVia^®^ VB40). The virtual non-calcium (VNCa) applications were used to assess bone marrow edema (BME). Color-coded maps superimposed on gray-scale CT images were available in the postprocessing software application. In particular, on 3D imaging, a green-blue-scale was employed, coding the normal bone in blue and the edema in green. In a colored-scale, the 2D images were orientated in the different planes, and edema was coded in green to yellow or in red, according to the cut-off chosen and the type of color-coding adopted (rainbow or violet-green). For each participant, the isotropic image dataset was analyzed using a soft tissue and bone window on the preferred imaging planes. 

### 2.5. Image Analysis

MRI and DECT were analyzed by four independent (R1 to R4) radiologists (GF, EO, CL, and VR, with 15, 11, 4, and 2 years of experience, respectively). The images were analyzed in a random order during three reading sessions separated by 1-month intervals. The readers were blinded to the clinical/microbiological findings. A diagnosis of osteomyelitis was retained based on a comprehensive assessment that included imaging, clinical, and microbiological (bone biopsy or surgery) features.

At MRI, the T1 signal intensity was considered abnormal if the signal of the affected bone marrow had decreased compared with the normal adjacent fatty marrow, while BME was evaluated in fluid-sensitive images (fat-suppressed T2W/proton density-weighted and STIR). Contrast-enhanced T1 imaging was used to confirm the diagnosis, to better delineate the bone abscess or infarction, and to better identify the associated findings in the surroundings tissues, including abscesses or fistulous tracts.

At DECT, the diagnosis of osteomyelitis was defined by the presence of BME of the affected bone segment on color-coded VNCa imaging, with the presence of bone erosion (cortical thinning with blurred margins) or bone resorption around the oedema if far from the cortical bone. In addition, any associated soft tissue findings, including the presence of abscesses or fistulous tracts, with or without gastric elements, were used to corroborate the diagnosis.

For each case and for each imaging tool, the four radiologists rated the patient disease status as follows: 1 = presence of osteomyelitis; 2 = probably osteomyelitis; 3 = non-specific findings; 4 = probably no osteomyelitis; or 5 = no osteomyelitis.

### 2.6. Clinical Findings and Microbiological Analysis 

All patients were evaluated at our Infectious Disease department and a daily clinical examination was ordered for each of them during the time of hospitalization. Clinical characteristics such as local pan, swelling or redness, fever, and functional limitations were evaluated. The demographic characteristics were also registered. 

During the hospitalization period, patients underwent either a bone biopsy or surgery for debridement or amputation. The sampled material was analyzed by our Microbiological department.

### 2.7. Statistical Analysis

The statistical analysis was conducted with STATA software vers. 16 (StataCorp. 2019. Stata Statistical Software: Release 16. College Station, TX, USA: StataCorp LLC) by an experienced statistical analyst (CM, 5 years of experience). 

Data available from a previous study performed at our institution, with a similar research design and methodology and similar objectives/hypotheses, were used to define the sample size. 

Demographic and clinical data were summarized using descriptive statistics and measures of variability. All parameters were reported with 95% confidence intervals. The statistical models and estimations were adjusted for covariates when necessary. A multi-reader multi-case analysis of the variance was performed using the MRMCoav R package (Version 0.1.3). The sensitivity and specificity were calculated using a binary threshold (score 1 and 2 = osteomyelitis; score 3, 4, or 5 = no presence of osteomyelitis). The multi-observer agreement was calculated by Cohen’s K index. Any disagreement was resolved by a consensus review between the specialists.

To further detail the methods, the summary call produces the ANOVA results for a global test of equality of the ROC AUC means across all imaging modalities and tests of pairwise differences, along with the confidence intervals for the differences and intervals for the individual modalities. 

The sensitivity, specificity, and ROC areas were calculated for each parameter, and comparisons between the AUCs were performed. Clinical, microbiological, and MRI findings were set as the reference standard for diagnosis. 

We also analyzed the diagnostic performance of DECT in the identification of BME alone, given MRI results as the standard of reference.

A multivariable logistic model was used to predict the probability of osteomyelitis, adjusted for the following regressors: age and sex. A *p* value of <0.05 was considered statistically significant.

## 3. Results 

### 3.1. Participant Characteristics

In total, 67 patients with suspected osteomyelitis were identified. Twenty-three participants were excluded for the following reasons: lack of CT (*n* = 15), unsuitable candidates for microbiological confirmation of osteomyelitis (*n* = 6), and MRI motion artifacts (*n* = 2). The final study sample was composed of 44 participants reporting osteomyelitis (mean age 62.5 years ± 16.5 [SD]), of which there were 32 men (73%) and 12 women (27%). The participants’ clinical data are summarized in Table 1, while a flowchart briefly outlining the participant outcomes shown in Figure 1. 

Out of the 44 participants evaluated for the presence of osteomyelitis, 32 (73%) were subsequently confirmed clinically, microbiologically, and at MRI imaging, while 12 (27%) were found to have no signs of osteomyelitis at presentation. For all the cases, there were different segments analyzed, in particular: 30 foots, 9 knee-legs, and 5 hips.

### 3.2. Clinical and Microbiological Results

Overall, clinical and microbiological data revealed osteomyelitis in 32 of the 44 participants (73%). Among the remaining 12 participants (27%) for which no osteomyelitis was confirmed, other diagnoses were obtained instead: cellulitis = 7; neuropathic arthropathy = 3; fasciitis = 1; osteonecrosis = 1.

### 3.3. Imaging Results

The sensitivity and specificity values, AUCs, and delta mean values for the diagnosis of osteomyelitis are reported in Table 2, Table 3 and Table 4. Figure 2 summarizes the ANOVA results for multi-reader multi-case analysis of ROC AUC across all imaging modalities.

DECT showed a good overall performance with respect to MRI. For DECT, the mean sensitivity and specificity were 89.0% (82.2, 93.8) and 72.9% (58.2, 84.7), while for MRI they were 89.1% (82.3, 93.9) and 87.5% (74.8, 95.3), respectively. For DECT, the mean positive and negative predictive values were 89.7% (83.0, 94.4) and 71.4% (56.7, 83.4), while for MRI they were 95.0% (89.4, 98.1) and 75.0% (61.6, 85.6), respectively. For what pertains to the overall performance in diagnosing osteomyelitis, the mean AUC values (as averaged from the four readers) resulted higher for MRI (AUC = 0.88) compared with DECT (AUC = 0.81), with a statistically non-significant difference (*p* = 0.12). 

When considering each imaging finding alone, the best accuracy was achieved when considering BMEs (AUC for DECT 0.85 versus AUC of MRI of 0.93, with *p* = 0.07), followed by the presence of bone erosions (AUC 0.77 for DECT and 0.53 for MRI, with *p* = 0.02), abscesses (AUC 0.42 for DECT versus 0.39 for MRI, with *p* = 0.43), sinus tracts (AUC.42 for DECT versus 0.41 for MRI, with *p* = 0.45) and surrounding gaseous elements (AUC 0.48 for DECT versus 0.45 for MRI, with *p* = 0.38).

Concerning the identification of BME alone, using MRI as the reference for diagnosis, DECT had a sensitivity of 91.2% (84.8, 95.5), a specificity of 82.4% (69.1, 91.6), a positive predictive value of 92.7% (86.6, 96.6), and a negative predictive value of 79.2% (65.9, 89.2).

### 3.4. Inter-Observer Agreement

For osteomyelitis analysis, a very high inter-reader agreement was achieved for DECT (k = 88; 95% CI: 77, 96) and MRI (k = 90; 95% CI: 78, 98).

Figure 3 and Figure 4 show example images of participants for which at least one reader missed diagnosing osteomyelitis.

### 3.5. Multivariable Logistic Analysis

In multivariable analysis, the diagnosis of osteomyelitis was unrelated to age or sex (*p* > 0.05 for all). 

## 4. Discussion

### 4.1. Discussion of Background

The DECT reports for the diagnosis of osteomyelitis, made using multiple imaging parameters, were compared with the MRI reports, with evidence of a similar diagnostic accuracy of osteomyelitis in the lower limb as with MRI (*p* = 0.12).

Osteomyelitis is associated with high morbidity and high healthcare costs and may require aggressive surgery or amputation. Thus, the prompt diagnosis is of great importance in guiding appropriate medical and surgical treatments. MRI is the most reliable imaging tool for the evaluation of osteomyelitis due to its high sensitivity and specificity performance. At MRI, STIR hyperintensity with corresponding T1-weighted hypointensity characterizes the typical bone marrow signaling abnormalities. However, false positive cases were described because of the confluent intramedullary patterns in T1W images. In addition, normal T1 signals have been reported in skeletal segments with confirmed osteomyelitis at the pathology examination, possibly reflecting a necrotic bone with fatty marrow. In this clinical setting, the associated anatomical findings, including ulcers, abscesses, and bone erosions, can play a crucial role to help the diagnosis. 

### 4.2. Role of DECT and Comparison with Previous Studies

DECT has been successfully used to identify BMEs in traumatic and non-traumatic settings. Furthermore, DECT has been applied for the evaluation of several districts, including the hip, the knee, and the foot. One of the major strength points of DECT is its ability to yield high-resolution isotropic images that can also be reconstructed with the bone of soft tissue kernel. These images may help in highlighting fine anatomic details both in the bone and adjacent tissues.

In a recent study evaluating DECT for diagnosing osteomyelitis, among 26 positive cases, the sensitivity ranged from 53.8% (by reading VNCa images alone) to 80.8% with the combined use of VNCa and standard CT images [23]. In the above-mentioned study, the specificity values ranged from 84.9% to 71.2% and decreased when evaluating both VNCa and standard images together. However, most patients were negative according to the reference standard; the reason for the decreased accuracy in the reading of both BME maps and standard images may be attributable in the inclusion of several different bone segments in both the upper and lower limbs. In clinical practice, the shape of the bone, the presence of cortical bone thickening, or the presence of reduced mineral density may interfere with the diagnosis of BME [24]. 

Our results confirm that DECT is accurate in diagnosing osteomyelitis of the lower limb, yielding similar accuracy values with respect to MRI, with an overall AUC of 0.81, and with 89.0% and 72.9% sensitivity and specificity, respectively. To achieve a higher statistical validity, we performed a multi-reader multi-case analysis. Overall, the results were similar among the four radiologists, despite the different experiences of the readers. Such agreement confirms that DECT may be considered a reliable imaging tool in this setting. 

In our opinion, and in accordance with other recently published reports, the evaluation of both BME maps and standard high-resolution images is essential for a correct osteomyelitis diagnosis [23].

Thanks to DECT color-coded images, it is possible to identify the presence of BME, a nonspecific marker of chronic pain, and one of the key imaging findings for the diagnosis of osteomyelitis in the lower limb [19,20,25]. In clinical practice, BME, as detected with an MRI, is used for the identification and demarcation of osteomyelitis foci [10,26,27,28]. 

In this study, DECT correctly identified the presence of BMEs in most patients, with an average sensitivity and specificity of 91.2% and 82.4%, (AUC = 0.90), with respect to MRI. These values are in line with those previously reported for other skeletal segments (ankle and knee) [22,29,30].

### 4.3. Discussion of Specific Imaging Parameters

On the other hand, BME is a nonspecific marker of bone damage and can be detected in other pathological conditions. For this reason, other imaging parameters evaluated on standard high-resolution CT images, both with bone and soft tissue windows, can play a crucial role to achieve diagnosis, and, above all, to rule out other differentials (such as stress fractures or osteo-chondral lesions). In our study, the parameter “bone erosions at the site of infection” allowed the diagnosis of the presence of osteomyelitis only in 8/44 cases but helped to increase the diagnostic likelihood and reduce the inter-observer agreement. Of fundamental importance, when bone erosions are not present, the diagnosis of osteomyelitis is improbable (specificity of 79.2% with NPV of 53.5%). 

On the other hand, all other imaging parameters, when considered alone, achieved diagnostic accuracy values that are inferior to those of BME. Among these parameters, gaseous elements can represent a key finding; when present, these are typically associated with infection and can be clearly depicted on CT. However, gaseous elements were present only in six patients in our series. Fistulous tracts and soft tissue abscesses represent other important imaging findings that may help radiologists in this setting. As a matter of fact, when using a non-contrast scan, thanks to the improved soft tissues contrast resolution, DECT can be applied to identify soft tissue’s involvement [31]. Nevertheless, for the evaluation of these parameters, DECT is still of limited value when compared with the contrast-enhanced MRI. Even if an additional contrast-enhanced CT scan could raise the detection rate of abscesses and fistulous tracts, and possibly ameliorate the overall accuracy of CT, it is our belief that acquiring an unenhanced CT represents one of the major strengths of DECT itself. 

### 4.4. Strenghts and Drawbacks of DECT

While MRIs still represents the reference standard for diagnosis, DECTs may play a role in patients unable to undergo an MRI, or when am MRI is not readily available. Numerous patients with suspect osteomyelitis may suffer from chronic invalidating diseases (diabetes, vascular stenosis, neuropathy), and renal impairment might represent a limitation for both CT and MR contrast-enhanced studies. Conversely, DECT is readily available and the scanning time is very short, reducing the potential concerns regarding motion artifacts. Moreover, additional features of DECT may be useful for differential diagnoses, such as in the identification of gout or thanks to the reduction in peri-prosthetic artifacts in patients with metal hardware [32,33,34]. Our study population did not include any patients with metal hardware so that a comparison of MRI versus DECT in reducing metal artifacts was not carried out. However, in our experience, the detection of BME may be limited by the presence of metal hardware, both at MRI and DECT imaging. On the other hand, in the subgroup of patients enrolled in this study who also had their hardware previously removed, metal-induced artifacts were better controlled on DECT than on MRI.

Furthermore, it should be underlined that morphological changes in the bone and soft tissues can be easily identified on the standard CT. As a matter of fact, the possible use of CT for diagnosing osteomyelitis has been proposed in previous studies that compared MRI and CT; Chandnani et al. showed a lower performance of single-energy CT when compared to MRI [15]. In another study by Gold et al., CT was useful in the detection of sequestrum, while MR imaging was helpful in defining the extent of the inflammatory process and in distinguishing osteomyelitis from cellulitis [35]. However, technological advancements allowed significant improvements for CT imaging [15,35]. Actually, in a more recent systematic review exploring the capacity of imaging tests to diagnose osteomyelitis, 81 studies were considered, showing that single-energy CT has a sensitivity of 69.7% and a specificity of 90.2% in detecting osteomyelitis [36]. 

The possibility of identifying BME represents an important additional feature of DECT [28,37]. However, bone marrow edema itself can be misleading as non-specific and can be observed in association with other non-infectious phenomena [38]. For this reason, we coupled morphological parameters and our results showed that the coexistence of multiple findings in the same patient can increase the diagnostic accuracy. 

Nonetheless, DECT may be associated with a higher radiation exposure. In our series, patients were relatively old (62.5 yo). In spite of that, thanks to the use of a software for dose modulation, and the acquisition of a single scan of the lower limb, the radiation burden should not represent a major concern in our population. Conversely, MRI should be preferred for young patients, especially if repeated checks could be expected in the management and follow-up. 

### 4.5. Limitations and Conclusions

Our study has some limitations. Firstly, we enrolled a relatively limited number of patients, although previous studies comparing DECT and MRI were carried out, on average, on comparable numbers. Secondly, we performed only a qualitative assessment of BME and associated imaging parameters. Furthermore, we did not perform a quantitative assessment of DECT numbers in the areas of BME to avoid the use of different cut-offs for the diagnosis of BME in the different segments analyzed. Finally, different segments were assessed in this study, namely the hip, knee, leg, and foot, potentially representing a source of confounding variables. However, the imaging appearance of osteomyelitis was similar across these districts, and we registered a higher prevalence of feet localization in comparison to other segments. Other limitations include the lack of specific imaging-clinical correlation and the inability to rule out the presence of overlapping diseases associated with BME, such as stress fractures or BME syndromes.

In conclusion, DECT showed a high diagnostic performance in the diagnosis of osteomyelitis. In our view, DECT might represent a useful alternative for the diagnosis of osteomyelitis when MRI is not available, or patients have contraindications to its execution. 

## Figures and Tables

**Figure 1 diagnostics-13-00703-f001:**
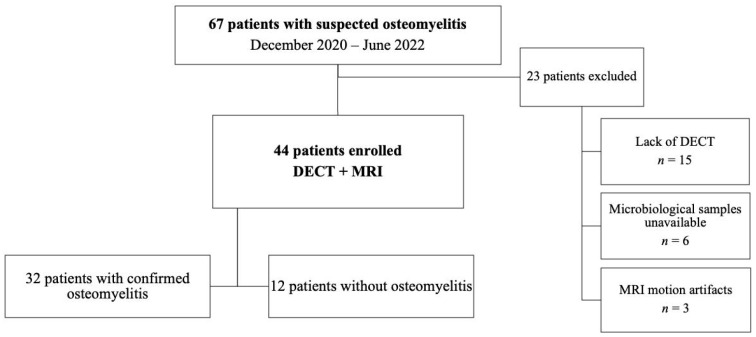
Flowchart of participants. DECT = dual-energy CT; MRI = magnetic resonance imaging.

**Figure 2 diagnostics-13-00703-f002:**
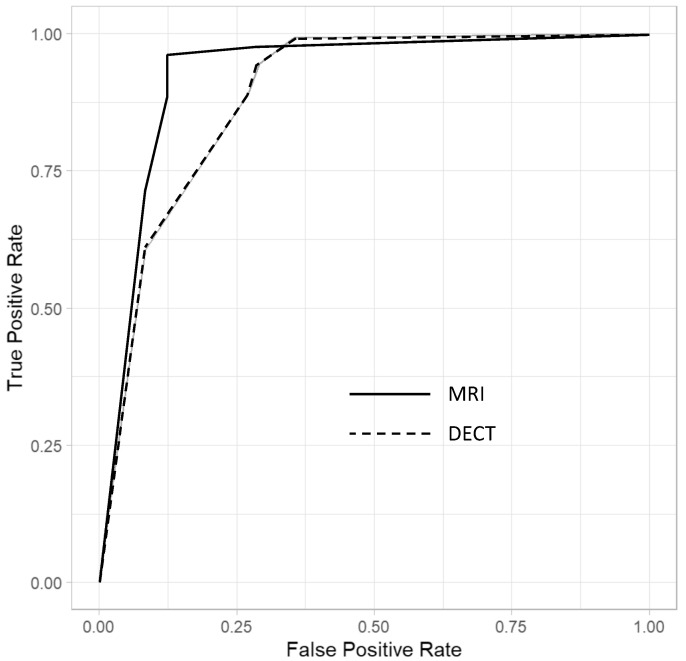
Diagnostic performance of the multi-reader multi-case analysis across all imaging modalities for MRI and DECT. The dashed line-curve represents DECT and the continuous line-curve represents MRI.

**Figure 3 diagnostics-13-00703-f003:**
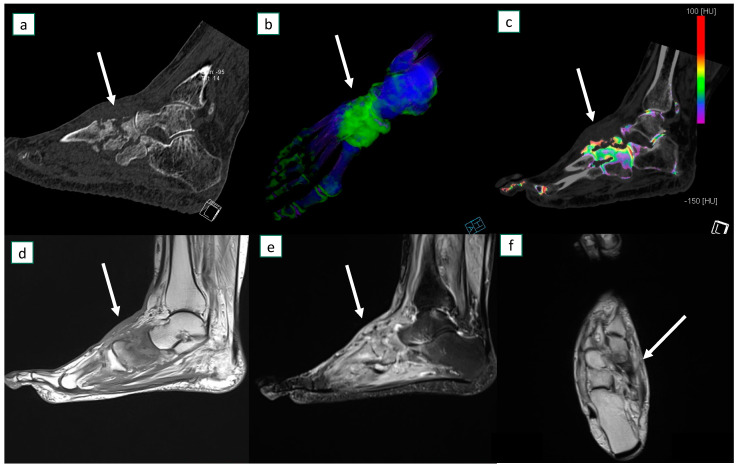
A 45-year-old female presented with acute atraumatic left foot pain and was diagnosed with osteomyelitis at bone biopsy. At dual-energy CT in sagittal plane there is evidence of advanced cortical erosions (**a**). At bone marrow edema analysis on 3D map (**b**) and sagittal 2D reconstruction image (**c**), diffuse bone marrow edema (arrows) can be identified around the midfoot. MRI sagittal T1 sequence (**d**), STIR sagittal (**e**), and T1 axial (**f**) images confirm the corresponding diffuse marrow edema (arrows) and extensive soft tissue edema.

**Figure 4 diagnostics-13-00703-f004:**
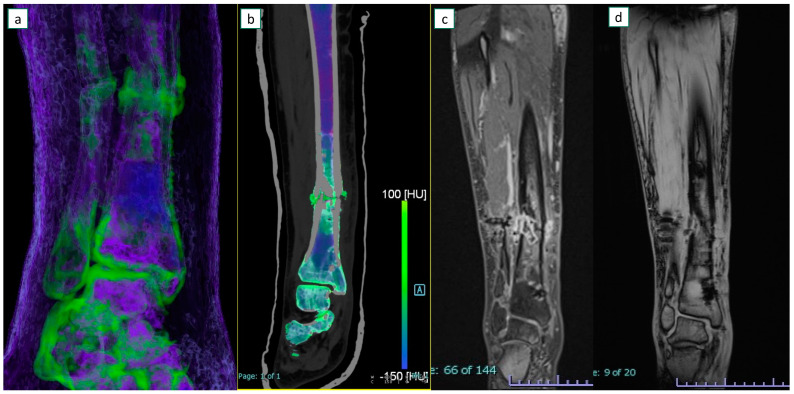
A 52-year-old female with osteomyelitis on tibial exposed fracture. On coronal 3D (**a**) and 2D (**b**) dual-energy CT images, severe bone marrow edema is identified around the fracture line. Additional edema is also present on the adjacent soft tissues. On the corresponding coronal MRI T1-weighted image after contrast material administration (**c**), additional edema with a sequestrum is depicted around the fracture foci. The corresponding T1-weighted image without contrast material (**d**) was considered non-diagnostic because of some metal-induced artifacts from a previous surgical fixation.

**Table 1 diagnostics-13-00703-t001:** Demographic and clinical data of the participants.

Characteristic	No. of Participants (*n* = 44)
Age (y)	62.5 (18–87) [16.5]
Number of men	32 (72.7%)
Number of women	12 (27.3%)
Osteomyelitis	32 (72.7%)
No osteomyelitis	12 (27.3%)
Side	
Right	21 (47.7%)
Left	23 (52.3%)
Skeletal segment	
Foot	30 (68.1%)
Knee/leg	9 (20.5%)
Hip	5 (11.4%)

Note—Demographic data are presented as mean values with (range) and [standard deviation]. Clinical data are presented as the number of cases with (percentage).

**Table 2 diagnostics-13-00703-t002:** Diagnostic performance of four readers at DECT versus MRI for diagnosing presence of osteomyelitis.

	Sensitivity	Specificity	AUC	PPV	NPV
MRI	89.1% * [82.3, 93.9]	87.5% * [71.8, 95.3]	0.88 * [0.78, 0.98]	95.0% [89.4, 98.1]	75.0% [61.6, 85.5]
Reader 1	93.8% [79.2, 99.2]	91.7% [61.5, 99.8]	0.93 [0.84, 1.00]	96.8% [83.3, 99.9]	84.6% [54.6, 98.1]
Reader 2	84.4% [67.2, 94.7]	91.7% [61.5, 99.8]	0.88 [0.78, 0.98]	96.4% [81.7, 99.9]	68.8% [41.3, 89.0]
Reader 3	87.5% [71.0, 96.5]	83.3% [51.6, 97.9]	0.85 [0.73, 0.98]	93.3% [77.9, 99.2]	71.4% [41.9, 91.6]
Reader 4	90.6% [75.0, 98.0]	83.3% [51.6, 97.9]	0.87 [0.75, 0.99]	93.5% [78.6, 99.2]	76.9% [46.2, 95.0]
DECT	89.0% * [82.2, 93.8]	72.9% * [58.2, 84.7]	0.81 * [0.67, 0.95]	89.7% [83.0, 94.4]	71.4% [56.7, 83.4]
Reader 1	93.8% [79.2, 99.2]	75.0% [42.8, 94.5]	0.84 [0.71, 0.98]	90.9% [75.7, 97.7]	81.8% [48.2, 97.7]
Reader 2	84.4% [67.2, 94.7]	75.0% [42.8, 94.5]	0.80 [0.65, 0.94]	90.0% [73.5, 97.9]	64.3% [35.1, 87.2]
Reader 3	93.8% [79.2, 99.2]	66.7% [34.9, 90.1]	0.80 [0.66, 0.95]	88.2% [72.5, 96.7]	80.0% [44.4, 97.5]
Reader 4	81.2% [63.6, 92.8]	75.0% [42.8, 94.5]	0.78 [0.64, 0.93]	89.7% [72.6, 97.8]	60.0% [32.2, 83.7]

Note—Percentages; fraction; 95% CIs in brackets. MRI = magnetic resonance imaging, DECT = dual-energy CT, AUC = area under the receiver operating characteristic (ROC) curve, PPV = positive predictive value, NPV = negative predictive value. * Mean sensitivity, specificity, and AUC of the four readers.

**Table 3 diagnostics-13-00703-t003:** Diagnostic performance of four readers at DECT for diagnosing presence of bone marrow edema, giving MRI as gold standard.

	Sensitivity	Specificity	AUC	PPV	NPV
DECT	91.2% * [84.8, 95.5]	82.4% * [69.1, 91.6]	90 * [0.84, 0.96]	92.7% [86.6, 96.6]	79.2% [65.9, 89.2]
Reader 1	90.6% [75.0, 98.0]	83.3% [51.6, 97.9]	0.87 [0.75, 0.99]	93.5% [78.6, 99.2]	76.9% [46.2, 95.0]
Reader 2	87.1% [77, 96]	84% [71, 94]	0.87 [0.80, 0.94]	96.8% [83.3, 99.9]	84.6% [54.6, 98.1]
Reader 3	85% [72, 94]	96% [85, 100]	0.90 [0.84, 0.96]	96.8% [83.3, 99.9]	84.6% [54.6, 98.1]
Reader 4	87% [74, 95]	91% [79, 98]	0.89 [0.83, 0.96]	96.8% [83.3, 99.9]	84.6% [54.6, 98.1]

Note—Percentages; fraction; 95% CIs in brackets. MRI = magnetic resonance imaging, DECT = dual-energy CT, AUC = area under the receiver operating characteristic (ROC) curve, PPV = positive predictive value, NPV = negative predictive value. * Mean sensitivity, specificity, and AUC of the four readers.

**Table 4 diagnostics-13-00703-t004:** Diagnostic performance of four readers at DECT versus MRI for diagnosing presence of BMEs, erosions, abscesses, sinus tracts, and gas.

	MRI			DECT		
	Sensitivity	Specificity	AUC	Sensitivity	Specificity	AUC
BME	94.5% * [79.5, 99.6]	91.6% * [61.5, 99.8]	0.93 *	89.1% * [74.5, 97.8]	81.2% * [50.3, 96.8]	0.85 *
Reader 1	93.8% [79.2, 99.2]	100% [73.5, 1.00]	0.97 [0.93, 1.00]	90.6% [75.0, 98.0]	83.3% [51.6, 97.7]	0.87 [0.75, 0.99]
Reader 2	93.8% [79.2, 99.2]	91.7% [61.5, 99.8]	0.93 [0.81, 0.97]	87.5% [71.0, 96.5]	83.3% [51.6, 97.9]	0.85 [0.73, 0.98]
Reader 3	93.8% [79.2, 99.2]	91.7% [61.5, 99.8]	0.93 [0.81, 0.97]	90.6% [75.0, 98.0]	75.0% [42.8, 94.5]	0.82 [0.69, 0.97]
Reader 4	96.9% [83.8, 99.9]	83.3% [51.6, 97.9]	0.90 [0.79, 1.00]	87.5% [71.0, 96.5]	83.3% [51.6, 97.7]	0.85 [0.73, 0.98]
Erosions	49.2% * [40.3, 58.2]	89.4% * [77.3, 96.5]	0.53 *	74.2% * [65.7, 81.5]	79.2% * [65.0, 89.5]	0.77 *
Reader 1	34.4% [18.6, 53.2]	91.7% [61.5, 99.8]	0.37 [0.25, 0.49]	68.8% [50.0, 83.9]	83.3% [51.6, 97.7]	0.76 [0.62, 0.90]
Reader 2	46.9% [29.1, 65.3]	91.7% [61.5, 99.8]	0.31 [0.19, 0.43]	75.0% [56.6, 88.5]	83.3% [51.6, 97.9]	0.79 [0.65, 0.93]
Reader 3	56.3% [37.7, 73.6]	83.3% [51.6, 97.9]	0.70 [0.56, 0.84]	78.1% [60.0, 90.7]	66.7% [34.9, 90.1]	0.72 [0.57, 0.88]
Reader 4	59.4% [40.6, 76.3]	91.7% [61.5, 99.8]	0.75 [0.63, 0.87]	75.0% [56.6, 88.5]	83.3% [51.6, 97.7]	0.79 [0.65, 0.93]
Abscesses	31.2% * [23.4, 40.4]	91.7% * [80.0, 97.7]	0.39 *	26.6% * [19.1, 35.1]	89.6% * [77.3, 96.5]	0.42 *
Reader 1	31.2% [16.1, 50.0]	75.0% [42.8, 94.5]	0.47 [0.31, 0.62]	25.0% [11.5, 43.4]	83.3% [51.6, 97.9]	0.46 [0.32, 0.59]
Reader 2	43.8% [26.4, 62.3]	91.7% [61.5, 99.8]	0.32 [0.20, 0.44]	34.4% [18.6, 53.2]	91.7% [61.5, 99.8]	0.37 [0.25, 0.49]
Reader 3	28.1% [13.7, 46.7]	100% [73.5, 100]	0.36 [0.28, 0.44]	25.0% [11.5, 43.4]	91.7% [61.5, 99.8]	0.42 [0.30, 0.53]
Reader 4	21.9% [9.3, 40.0]	100% [71.5, 100]	0.39 [0.32, 0.46]	21.9% [9.3, 40.0]	91.7% [61.5, 99.8]	0.43 [0.32, 0.54]
Sinus tract	22.7% * [15.7, 30.9]	95.8% * [85.7, 99.5]	0.41 *	21.9% * [15.1, 30.0]	93.8% * [82.8, 98.7]	0.42 *
Reader 1	28.1% [13.7, 46.7]	91.7% [61.5, 99.8]	0.40 [0.29, 0.51]	25.0% [11.5, 43.4]	91.7% [61.5, 99.8]	0.42 [0.31, 0.53]
Reader 2	25.0% [11.5, 43.4]	91.7% [61.5, 99.8]	0.41 [0.31, 0.53]	28.1% [13.7, 46.7]	91.7% [61.5, 99.8]	0.40 [0.29, 0.51]
Reader 3	25.0% [11.5, 43.4]	100% [73.5, 100]	0.38 [0.30, 0.45]	18.8% [7.2, 36.4]	91.7% [61.5, 99.8]	0.45 [0.34, 0.55]
Reader 4	12.5% [3.5, 29.0]	100% [73.5, 100]	0.44 [0.38, 0.50]	15.6% [5.3, 32.8]	100% [73.5, 100]	0.42 [0.36, 0.49]
Gas	12.5% * [7.3, 19.5]	97.9% * [65.9, 89.2]	0.45 *	15.7% * [9.8, 23.1]	89.6% * [77.3, 96.5]	0.48 *
Reader 1	9.4% [2.0, 25.0]	91.7% [61.5, 99.8]	0.49 [0.40, 0.59]	9.4% [2.0, 25.0]	91.7% [61.5, 99.8]	0.49 [0.40, 0.59]
Reader 2	18.8% [7.2, 36.4]	100% [73.5, 100]	0.40 [0.34, 0.47]	18.8% [7.2, 36.4]	91.7% [61.5, 99.8]	0.45 [0.34, 0.55]
Reader 3	9.4% [2.0, 25.0]	100% [73.5, 100]	0.45 [0.40, 0.50]	15.6% [5.3, 32.8]	83.3% [51.6, 97.9]	0.51 [0.38, 0.63]
Reader 4	12.5% [3.5, 29.0]	100% [73.5, 100]	0.44 [0.38, 0.50]	18.8% [7.2, 36.4]	91.7% [61.5, 99.8]	0.45 [0.34, 0.55]

Note—Percentages; 95% CIs in brackets. MRI = magnetic resonance imaging, DECT = dual-energy CT, AUC = area under the receiver operating characteristic (ROC) curve, BME = bone marrow edema. * Mean sensitivity, specificity, and AUC of the four readers.

## Data Availability

Additional data are unavailable due to privacy or ethical restrictions.

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
