# Peer review of "Osteomyelitis of the Lower Limb: Diagnostic Accuracy of Dual-Energy CT versus MRI"

_diagnostics, 2023, doi:10.3390/diagnostics13040703_

Round 1
Reviewer 1 Report
Dear authors,
In this study, you sought to assess the diagnostic performance of DECT vs MRI for osteomyelitis.
Overall, this is a well-conducted paper.
However, I would like to draw authors’ attention to the fact that there are some language mistakes in the article that need to be corrected before considering publication.
Please find examples below:
'Foot osteomyelitis is the segment most often interested in diabetic patients.' Note that the selection of the word ‘interested’ is not appropriate here.
‘Finally, we assessed hip, knee, leg and foot, potentially representing a bias.’ This sentence should be re-phrased please.
Therefore, I would advise language polishing ideally performed by a native English speaker.
On top of that, although the content of the discussion section is interesting, I would advise you to organize this section more appropriately. To be more exact, the first (ie opening) paragraph is too short and you also start discussing recent studies prematurely.
I would also advise you to add subtitles to the discussion section so that you facilitate reading.
From a scientific point of view, I would be interested to read a quick discussion on ionizing radiation issues secondary to DECT.
Author Response
Dear authors,
In this study, you sought to assess the diagnostic performance of DECT vs MRI for osteomyelitis.
Overall, this is a well-conducted paper.
However, I would like to draw authors’ attention to the fact that there are some language mistakes in the article that need to be corrected before considering publication.
Please find examples below:
'Foot osteomyelitis is the segment most often interested in diabetic patients.' Note that the selection of the word ‘interested’ is not appropriate here.
GF: interested was changed with involved
All the manuscript was checked for spelling and English edited, as suggested.
‘Finally, we assessed hip, knee, leg and foot, potentially representing a bias.’ This sentence should be re-phrased please.
GF Thank you for the comments; the sentence was changed as follows:
“Finally, different segments were assessed in this study, namely hip, knee, leg and foot, potentially representing a bias. However, the imaging appearance of osteomyelitis was similar in these districts, and we had a prevalence of foot in comparison to other segments.”
Therefore, I would advise language polishing ideally performed by a native English speaker.
GF All the manuscript was checked for spelling and English edited, as suggested.
On top of that, although the content of the discussion section is interesting, I would advise you to organize this section more appropriately. To be more exact, the first (ie opening) paragraph is too short and you also start discussing recent studies prematurely.
I would also advise you to add subtitles to the discussion section so that you facilitate reading.
From a scientific point of view, I would be interested to read a quick discussion on ionizing radiation issues secondary to DECT.
GF thanks for the suggestions.
As indicated subtitles were added to the discussion section to facilitate reading.
Also, a quick discussion on ionizing radiation issues secondary to DECT was included in discussion section, as suggested.
Reviewer 2 Report
This study aims to compare the diagnostic performance between DECT and MRI to detect and diagnose osteomyelitis. It could be of interest in clinical routine, but some concerns should be addressed before publication.
Dual Energy CT also has the limitation of limited access, which you mention as a barrier for MRI – are there other positive aspects of DECT over MRI? Especially e.g. the examination of patient with metal implants – you should discuss this further than the one sentence in the discussion.
You refer to contrast-enhanced MRI sequences, but no application of contrast material is described in the MRI imaging section – please add this information
Is the used Likert scale established? Please add the reference
Probability of osteomyelitis: why did you only analyze age and sex? What about underlying diseases or specific medication that increase the risk for osteomyelitis? Did you analyze these criteria?
What about the diagnostic accuracy regarding identification of differential diagnoses? You only analyzed patients with osteomyelitis, did you also correctly identify the patients that suffered from other diagnoses such as stress fractures?
Author Response
This study aims to compare the diagnostic performance between DECT and MRI to detect and diagnose osteomyelitis. It could be of interest in clinical routine, but some concerns should be addressed before publication.
Dual Energy CT also has the limitation of limited access, which you mention as a barrier for MRI – are there other positive aspects of DECT over MRI? Especially e.g. the examination of patient with metal implants – you should discuss this further than the one sentence in the discussion.
- As indicated a short additional paragraph concerning metal induced artifacats is now included in discussion
“Our study population did not include nay patient with metal hardware so that a comparison of MRI versus DECT in reducing metal artifacts was not carried out. However, in our experience, detection of BME may be limited by the presence of metal hardware both at MRI and DECT imaging. Conversely, in the subgroup of patients enrolled in this study, who had their hardware removed, metal induced artifacts were better controlled on DECT than on MRI.”
You refer to contrast-enhanced MRI sequences, but no application of contrast material is described in the MRI imaging section – please add this information
GF: the imaging parameters were included at the end of MR imaging section as suggested.
Is the used Likert scale established? Please add the reference
GF, We used a 5 point Likert scale. We have employed a similar approach in other papers and we were not asked to add the reference before. We have a very long reference list and we avoided to include references if not strictly related to the topic or needed to reproduce the specific study. Similarly for other statistical analysis we decided to avoid including the reference.
Probability of osteomyelitis: why did you only analyze age and sex? What about underlying diseases or specific medication that increase the risk for osteomyelitis? Did you analyze these criteria?
GF No we did not correlate the disease with other clinical data because: 1) data were available only for some of the enrolled patients; 2) we focused on imaging diagnosis and avoid to add too many clinical data.
What about the diagnostic accuracy regarding identification of differential diagnoses? You only analyzed patients with osteomyelitis, did you also correctly identify the patients that suffered from other diagnoses such as stress fractures?
- Patients were enrolled with the suspect and clinical indication of osteomyelitis. All patients diagnosis was confirmed by biopsy and clinical-lab data. We did not had other specific diagnosis. In the hypothesis of other overlapping diagnosis (including stress fractures, OCLs and transient BME syndromes) we still referred to MRI as standard for the diagnosis of BME.
A short paragraph was added at the end of limitations section.
“Other limitations included the lack of specific imaging-clinical correlation and the impossibility to rule out the presence of overlapping diseases associated with BME, such as stress fractures or BME syndromes.”